# Medical Data over Sound—CardiaWhisper Concept

**DOI:** 10.3390/s25154573

**Published:** 2025-07-24

**Authors:** Radovan Stojanović, Jovan Đurković, Mihailo Vukmirović, Blagoje Babić, Vesna Miranović, Andrej Škraba

**Affiliations:** 1Faculty of Electrical Engineering, University of Montenegro, 81000 Podgorica, Montenegro; 2MECOnet Ltd., 81000 Podgorica, Montenegro; jovan@meconet.me; 3Department of Cardiology, Faculty of Medicine, University of Montenegro, 81000 Podgorica, Montenegro; mihailov@ucg.ac.me (M.V.); blagoje.b@ucg.ac.me (B.B.); vesnami@ucg.ac.me (V.M.); 4Cardiology Clinic, Clinical Center of Montenegro, 81000 Podgorica, Montenegro; 5Institute for Children’s Disease, Clinical Center of Montenegro, 81000 Podgorica, Montenegro; 6Cybernetics & Decision Support Systems Laboratory, Faculty of Organizational Sciences, University of Maribor, Kidričeva cesta 55a, 4000 Kranj, Slovenia; andrej.skraba@um.si

**Keywords:** medical wearables, Data Over Sound (DoS), IoT in healthcare, edge computing, real-time signal processing, modulation and demodulation, JavaScript, near-ultrasound communication

## Abstract

Data over sound (DoS) is an established technique that has experienced a resurgence in recent years, finding applications in areas such as contactless payments, device pairing, authentication, presence detection, toys, and offline data transfer. This study introduces CardiaWhisper, a system that extends the DoS concept to the medical domain by using a medical data-over-sound (MDoS) framework. CardiaWhisper integrates wearable biomedical sensors with home care systems, edge or IoT gateways, and telemedical networks or cloud platforms. Using a transmitter device, vital signs such as ECG (electrocardiogram) signals, PPG (photoplethysmogram) signals, RR (respiratory rate), and ACC (acceleration/movement) are sensed, conditioned, encoded, and acoustically transmitted to a nearby receiver—typically a smartphone, tablet, or other gadget—and can be further relayed to edge and cloud infrastructures. As a case study, this paper presents the real-time transmission and processing of ECG signals. The transmitter integrates an ECG sensing module, an encoder (either a PLL-based FM modulator chip or a microcontroller), and a sound emitter in the form of a standard piezoelectric speaker. The receiver, in the form of a mobile phone, tablet, or desktop computer, captures the acoustic signal via its built-in microphone and executes software routines to decode the data. It then enables a range of control and visualization functions for both local and remote users. Emphasis is placed on describing the system architecture and its key components, as well as the software methodologies used for signal decoding on the receiver side, where several algorithms are implemented using open-source, platform-independent technologies, such as JavaScript, HTML, and CSS. While the main focus is on the transmission of analog data, digital data transmission is also illustrated. The CardiaWhisper system is evaluated across several performance parameters, including functionality, complexity, speed, noise immunity, power consumption, range, and cost-efficiency. Quantitative measurements of the signal-to-noise ratio (SNR) were performed in various realistic indoor scenarios, including different distances, obstacles, and noise environments. Preliminary results are presented, along with a discussion of design challenges, limitations, and feasible applications. Our experience demonstrates that CardiaWhisper provides a low-power, eco-friendly alternative to traditional RF or Bluetooth-based medical wearables in various applications.

## 1. Introduction

Health is humanity’s greatest wealth, and regular monitoring is essential for timely intervention and effective treatment. Today, a wide range of health parameters can be self-monitored using wearable medical devices, often referred to as healthcare wearables or self-meters. These include blood pressure monitors, glucose meters, temperature sensors, oxygen saturation monitors, smartwatches, Holter monitors, loop recorders, pulse and rhythm analyzers, fall detectors, and more. Recent advances in wearable sensing technologies have significantly enhanced the accuracy and functionality of these devices, bringing their performance closer to that of clinical instruments [1,2]. The healthcare wearables market is expanding rapidly and is projected to reach EUR 140 billion by 2030, with one in three adults expected to incorporate these devices into their daily lives. Furthermore, the COVID-19 pandemic has accelerated the adoption and integration of wearables within broader telemedicine networks [3,4].

Despite these advancements, several research and innovation challenges persist in the design of healthcare wearables. Key issues include sensor accuracy, noise reduction, advanced signal processing, reliable communication in both near-field and far-field scenarios, data compression and storage, power efficiency, feature extraction and classification, AI-based assistance, and seamless integration with telemedicine and edge computing systems [5,6]. Additional limitations include high costs for consumers in developing countries, setup difficulties, limited memory for long-term recordings, and compatibility issues across various mobile operating systems (Android, iOS, Windows). Despite the widespread use of wireless technologies such as Bluetooth, Wi-Fi, and ZigBee in wearable healthcare systems, these protocols face critical limitations in medical environments, including electromagnetic interference (EMI) with sensitive equipment and high power consumption [7,8]. Studies have shown that EMI can cause hazardous incidents in medical devices and disrupt deep brain stimulation systems, underscoring the need for alternative communication methods—such as near-ultrasound acoustic communication—that are both low-power and resistant to EMI.

Data over sound (DoS) is a well-established technique for encoding and transmitting data using audible or inaudible sound waves. Originating with the Morse telegraph, DoS has found applications in telecommunications, touch-tone systems (DTMF, dual-tone multi-frequency signaling), and modem communications. Sound-based communication is also widely used underwater, enabling data transfer between submarines and underwater robots over distances ranging from tens to hundreds of kilometers. Data can be transmitted acoustically through air, wires, water, or even solid objects [9,10,11,12]. Audible frequencies typically range from 20 Hz to 20 kHz, while inaudible communication utilizes near-ultrasonic and ultrasonic frequencies above 20 kHz. The basic operation of DoS involves three main steps: (1) encoding, where analog or digital data are converted into a sound signal; (2) transmission, where the sound is emitted by a speaker or other sound-wave generator; and (3) reception, where a microphone captures the sound and decodes it back into data. Various modulation techniques—such as frequency-shift keying (FSK), amplitude-shift keying (ASK), phase-shift keying (PSK), and orthogonal frequency-division multiplexing (OFDM)—are commonly employed, with modulation and demodulation typically performed in software.

Beyond historical and underwater uses, DoS is now employed in a variety of real-world scenarios, including contactless payments, device pairing (e.g., TVs, smart speakers), authentication and presence detection, marketing and retail (audio beacons), toys, and offline data transfer in environments lacking Bluetooth or Wi-Fi connectivity. DoS offers several advantages: it works on most devices equipped with a speaker and microphone, does not require Internet or Bluetooth, can operate through walls (depending on frequency), requires minimal setup, consumes little power, supports a wide range of platforms, enables secure and private data transfer, integrates seamlessly without special tools, and is scalable and robust across extreme environments. It also supports offline, peer-to-peer, and one-to-many communication.

However, DoS has certain limitations. Data rates are generally slower than Bluetooth or Wi-Fi; it is more susceptible to noise and interference and has a limited range and bandwidth; and transmission can be attenuated by certain building materials. In terms of health risks, electromagnetic (EM)-based wireless communication has been extensively studied and is generally considered safe at low exposure levels, though concerns about long-term RF exposure remain. In contrast, DoS does not emit electromagnetic radiation and is not associated with cancer risk, but it can cause annoyance, stress, or hearing discomfort if misused (e.g., at high volumes)—a concern that is minimized in near-field communication. Overall, current evidence suggests that EM-based wireless technologies carry a low-to-moderate health risk, while DoS remains a low-risk alternative [13,14].

In this study, we propose the “CardiaWhisper” concept, which extends the data over sound (DoS) paradigm into the field of medical data over sound (MDoS). While microphones, smart speakers, and other acoustic sensors are already used in medicine for applications such as monitoring breathing patterns, snoring, coughing, sneezing, fall detection, distress sounds, voice biomarkers, probe health, on-body communication, and near-field implant communication, the direct application of MDoS remains largely unexplored. We contend that MDoS has the potential to address several outstanding challenges in medical wearables, including local connectivity, gateway integration, reduced power consumption, low cost, compatibility with widely available consumer devices, and support for haptic interfaces and smart clothing.

Section 2 presents the architecture, operational principles, and signal processing concepts underlying CardiaWhisper. Section 3 provides preliminary test results, evaluating key parameters and discussing design challenges, limitations, and potential applications. The paper concludes with a summary and an overview of the relevant literature.

## 2. Methodology

Figure 1 illustrates the principle of medical signal transmission over sound. As a case study, the electrocardiogram (ECG) signal is used, captured via three electrodes in Einthoven’s triangle configuration placed on the patient’s chest. The signal is amplified, preprocessed, and transmitted by the CardiaWhisper transmitter (TX) device. Instead of relying on traditional electromagnetic radio waves, the TX employs a piezo speaker to transmit the ECG signal in the audio range of 16 kHz to 20 kHz. The receiver (RX), which may be a mobile phone, desktop computer, or tablet, captures the acoustic signal through its built-in microphone, processes the data, and can perform additional tasks such as visualization, decision support, logging, and networking.

Multiple receivers can simultaneously capture the transmitted signal, each using its own microphone. This capability ensures both redundancy and flexibility. For example, a doctor can view the data on a laptop, while a family member, the patient, or a nurse can monitor it on a mobile phone or tablet. The system design is scalable, allowing any number of compatible devices to join the network without the need for additional transmitters or specialized receivers. In addition, receivers can serve as gateways to edge computing platforms, other IoT devices, or data collectors. Local and remote staff, as well as the patient, can monitor the data in real time and take appropriate action as needed.

The hardware/software (HW/SW) architecture of the CardiaWhisper system is shown in Figure 2 illustrating the sequential chain of the sensor, transmitter, and receiver.

### 2.1. Transmitter

The CardiaWhisper transmitter, illustrated in Figure 2a, enables the transformation of signals as follows:$$\begin{array}{cc} \mathrm{ECG}(t) & \to V_{m}\left(t\right)\to \mathrm{Sound}\\ D[N..0] & \to V_{m}\left(t\right)\to \mathrm{Sound} \end{array}$$
where ECG(t) is the analog electrocardiogram signal, D[N..0] is a digital input, and $V_{m}\left(t\right)$ is the voltage-modulated output.

The transmitter consists of a signal amplifier (AMP), encoder (ENC), speaker driver (SD), speaker (SC), and power management (PM) module. AMP is a biomedical signal amplifier, specifically an ECG amplifier, which amplifies small electrical signals generated by the heart (typically 0.1 mV to 1 mV) and captured by electrodes. This results in a voltage signal ECG(t) at a level suitable for further processing (0 V to 5 V).

The signal ECG(t) is then encoded into a form suitable for transmission over air or wire. Analog and digital signals, such as a character D[N..0], a string, or any message, can be encoded for transmission. In the CardiaWhisper system, encoding is achieved through frequency modulation (FM), implemented either by a voltage-controlled oscillator (VCO) in a custom-designed chip or by a general-purpose microcontroller (MC).

The output from the VCO or MC is fed to the speaker driver (SD), which drives the piezo speaker (SC) with the FM-modulated signal $V_{m}\left(t\right)$. The instantaneous transmission frequency is given by(1)$$f_{m}\left(t\right)=f_{c1}+\left(\frac{f_{c2}-f_{c1}}{V_{\max}-V_{\min}}\right)\cdot \left(V\left(t\right)-V_{\min}\right)$$
where $f_{c1}$ and $f_{c2}$ are the frequencies corresponding to the minimum ($V_{\min}$) and maximum ($V_{\max}$) values of V(t), respectively. The frequency response of the speaker (SC) should be flat in the range $\left[f_{c1},f_{c2}\right]$.

A piezo speaker is selected for its low power consumption and high impedance. In our implementation, $f_{c1}$ and $f_{c2}$ correspond to the inaudible frequency range, specifically 16–20 kHz. For microcontroller-based implementations, FM modulation is achieved in software (firmware) by configuring built-in timers to operate in frequency generation mode. While MC-based modulation has lower resolution, it provides greater flexibility and simplifies the encoding of both analog and digital signals.

The power supply consists of a 9 V battery (BAT), with supply voltages $V_{\mathrm{CC}}=5\;V$ and $V_{\mathrm{CC}1}=9\;V$.

### 2.2. Receiver

In its basic configuration, the receiver (Figure 2b) consists of a microphone (MIC) and its amplifier, which are integral parts of the Audio Stack/Interface (AS) within any device, such as a mobile phone or tablet. The modules for acquisition, filtering, demodulation, and visualization are implemented in software. A microphone (MIC) captures the sound wave, in which the modulated ECG signal is embedded, and converts it into an electrical signal. This signal is then amplified and digitized by the AS, resulting in $V_{0}\left(kT_{s}\right)$, the digital equivalent of the transmitter’s analog signal $V_{m}\left(t\right)$. Here, $V_{0}\left(kT_{s}\right)$ represents a discrete-time vector of audio samples, commonly denoted as $V_{0}\left(n\right)$.

The sampling frequency of the AS is $F_{s}=\frac{1}{T_{s}}$, typically $F_{s}=44,100\;\mathrm{Hz}$ or $F_{s}=48,000\;\mathrm{Hz}$. In addition to the sampling circuit, the AS includes several additional components, such as gain control, a limiter, echo cancellation, and more. The resolution of the AS generally ranges from 16 to 24 bits. The signal $V_{0}\left(n\right)$ is then forwarded to the software signal processing block for further processing.

### 2.3. Signal Processing Approach

Here, we address a segment of signal processing algorithms, focusing primarily on software-based signal processing tasks performed on the receiver side (see Figure 2b), specifically those related to decoding medical signals embedded in sound. The illustration is implemented in MATLAB^®^ 24.1.0 [15], allowing for both methodological analysis and observation of effects.

The process of demodulating the $V_{0}\left(n\right)$ signal begins with its Band-Pass Filtering (BPF) within the range $\left[f_{\mathrm{low}},f_{\mathrm{high}}\right]=\left[f_{c1},f_{c2}\right]$. The BPF can be implemented in both the time and frequency domains. In the time domain, the BPF is realized as an *N*-th order IIR Butterworth filter, represented by the following difference equation: (2)$$\begin{array}{cc} V_{1}\left[n\right] & =-a_{1}V_{1}\left[n-1\right]-a_{2}V_{1}\left[n-2\right]-\ldots -a_{N}V_{1}\left[n-N\right]\\ \; & \;+b_{0}V_{0}\left[n\right]+b_{1}V_{0}\left[n-1\right]+b_{2}V_{0}\left[n-2\right]+\ldots +b_{N}V_{0}\left[n-N\right] \end{array}$$
where $a_{i}$ and $b_{i}$ are the feedback and feedforward coefficients, respectively, calculated from the filter design. An order of N=4 is sufficient for satisfactory filtering.

In the spectral domain, the BPF1 filter is implemented as(3)$$\begin{array}{c} V_{1}\left(t\right)=F^{-1}\left(S\left(f\right)\right)=F^{-1}\left(W\left(f\right)\cdot H\left(f\right)\right)=F^{-1}\left(F\left(V_{0}\left(n\right)\right)\cdot H\left(f\right)\right) \end{array}$$(4)$$\begin{array}{c} H\left(f\right)=\left\{\begin{array}{cc} 1, & f_{c1}\leq f\leq f_{c2},\\ 0, & \mathrm{otherwise}. \end{array}\right. \end{array}$$
where F and $F^{-1}$ are the operators of the forward and inverse Fast Fourier Transform (FFT), respectively; H(f) is the frequency response of the window filter; and (·) denotes element-wise multiplication (Hadamard product).

Figure 3 shows the time representation of the modulated signal $V_{0}\left(kT_{s}\right)$, its FFT spectrum, and the technique of spectrum-based filtering implemented with a window H(f).

Several FM demodulation (FM DEM) methods—in fact, estimators of instantaneous frequency—are tested in order to select the most appropriate for application in the CardiaWhisper system: the Hilbert transform-based estimator (HIL), implemented in versions HIL1 and HIL2; the derivative-based estimator (DIFF); and the zero crossing (ZC)-based estimator.

FM demodulation using the Hilbert transform is a technique that exploits the relationship between the phase and frequency of the FM signal. By extracting the instantaneous phase, differentiating it to obtain the frequency, and then recovering the message signal, this method provides a non-coherent and efficient way to demodulate FM signals.

Two implementations of Hilbert-transform-based FM demodulation were evaluated, denoted as HIL1 and HIL2, with MATLAB-style pseudocode provided below:

   **HIL1**

   y  = hilbert(V1(n));

   yd = y(2:end) .* conj(y(1:end-1));

   V2(n) = angle(yd);

   **HIL2**

   y  = hilbert(V1(n));

   yd = unwrap(angle(y));

   V2(n) = diff(yd) / (2*pi*(1/Fs));

Here, hilbert denotes the Hilbert transform operator, angle returns the phase of the complex analytic signal, and unwrap removes phase discontinuities. In the HIL1 equation, yd is obtained by element-wise multiplication of y(2:end) (the analytic signal starting at the second sample) with the complex conjugate of y(1:end−1). $F_{s}$ is the sampling frequency. $V_{2}\left(n\right)$ is then filtered by BPF2 to obtain the final demodulated signal:(5)$$\begin{array}{c} V^{\prime }\left(n\right)=\mathrm{BPF}2\left(V_{2}\left(n\right)\right) \end{array}$$
BPF2 is an *N*th-order Butterworth filter, as in Equation (Equation 2), and additionally performs DC blocking. Satisfactory results are obtained with N=2. The outputs of HIL1 and HIL2 are illustrated in Figure 3 as “blue” and “magenta” plots, respectively.

The derivative-based estimator computes the difference of the filtered signal $V_{1}\left(n\right)$ to produce an amplitude-modulated signal $V_{2}\left(n\right)$, which is then filtered with BPF2 to recover the original message:(6)$$\begin{array}{c} V^{\prime }\left(n\right)=\mathrm{BPF}2\left(V_{2}\left(n\right)\right)=\mathrm{BPF}2\left(\mathrm{DIFF}\left(V_{1}\left(n\right)\right)\right)=\mathrm{BPF}2(|V_{1}\left(n\right)-V_{1}\left(n-1\right)\left|\right) \end{array}$$

In our practical implementation, the band-pass filter (BPF1) used to isolate the modulated FM carrier within the 16–20 kHz band is realized as a digital infinite impulse response (IIR) Butterworth band-pass filter, implemented in the time domain.

Specifically, we use a 4th-order (or, in some tests, 6th-order) digital Butterworth band-pass filter, designed using the standard bilinear transform approach. The Butterworth design was chosen for its maximally flat frequency response in the passband and for moderate computational complexity, making it suitable for real-time processing in both MATLAB and JavaScript/Web Audio API environments. The filter is implemented as a cascade of biquad (second-order) sections for numerical stability, with the following specifications:Sampling Frequency: 48 kHz;Passband: 16–20 kHz.Filter Type: IIR Butterworth, order 4 or 6.Design Method: Bilinear transform with digital frequency pre-warping to preserve the sharpness of the band edges near the Nyquist frequency.Causality: The filter is fully causal, implemented in the standard direct form II transposed structure for efficiency.Zero-Phase Filtering: For offline MATLAB analysis (i.e., to demonstrate ideal performance), we sometimes apply zero-phase filtering using filtfilt to eliminate group delay; however, for real-time and browser-based applications, the filter is run in causal, single-pass mode.Transition Bands: With a 4 kHz wide passband at such a high frequency, filter order is a critical trade-off. We found that a 4th-order design provides acceptable attenuation (>30 dB) outside the 16–20 kHz band, but a 6th-order filter offers steeper roll-off if increased selectivity is required.Stability and Real-Time Suitability: The chosen implementation does not introduce instability or excessive group delay in the passband. Group delay at the upper edge (20 kHz) is moderate, and pre-detection of signal edges is not required for our application.

Given the close proximity of the upper band edge (20 kHz) to the Nyquist frequency (24 kHz for a 48 kHz sample rate), care was taken to avoid aliasing and to ensure that filter coefficients were designed with appropriate pre-warping.

Due to inherent limitations of digital filters operating at high frequencies near the Nyquist limit, practical implementations necessarily differ from theoretical models. Nevertheless, the implemented algorithm provides sufficient isolation of the modulated carrier to enable reliable demodulation with acceptable levels of distortion.

The demodulated signal obtained via the DIFF method is shown as the “green” plot in Figure 3.

In the case of the zero-crossing (ZC) estimator/discriminator, the time T(k) is defined as the difference between the current and previous zero crossing: (7)$$\begin{array}{c} T\left(k\right)=t_{\mathrm{zero}}\left(k\right)-t_{\mathrm{zero}}\left(k-1\right), \end{array}$$
where(8)$$\begin{array}{c} t_{\mathrm{zero}}\left(k\right)=\left\{\begin{array}{cc} k, & \mathrm{if}\;V_{1}\left(k\right)>0\;\mathrm{and}\;V_{1}\left(k-1\right)\leq 0,\\ k, & \mathrm{if}\;V_{1}\left(k-1\right)\geq 0\;\mathrm{and}\;V_{1}\left(k\right)<0. \end{array}\right. \end{array}$$

The value T(k) is then passed through the BPF2 filter to obtain the final demodulated signal: (9)$$\begin{array}{c} V^{\prime }\left(n\right)=\mathrm{BPF}2\left(V_{2}\left(n\right)\right)=\mathrm{BPF}2\left(c\cdot T\left(k\right)\right), \end{array}$$
where *c* is a proportional constant. The ZC-demodulated signal is illustrated in Figure 3 (red plot). As can be seen, the ZC technique is the simplest to implement while still providing satisfactory results. Figure 4 illustrates the demodulation processing steps for this method.

There are several additional methods for software demodulation of FM signals, such as demodulation using the FFT spectrum. As the FM signal varies, the maximum values (i.e., the highest peaks) in the FFT spectrum shift accordingly. The positions of these peaks are proportional to the instantaneous frequency of the FM signal. By tracking the peak frequency over time—specifically, by identifying the maximum value in each FFT frame—it is possible to reconstruct the instantaneous frequency.

Phase-Locked Loop (PLL)-based FM demodulation is another widely used technique for demodulating frequency-modulated signals, particularly due to its efficiency and simplicity. A PLL is a feedback control system that synchronizes the phase of a local oscillator with the phase of an incoming signal. In FM demodulation, the PLL tracks the instantaneous frequency of the FM signal, and the phase difference between the PLL’s oscillator and the incoming signal yields the message signal. The PLL can be implemented in either a coherent or non-coherent manner, depending on how the system is designed to synchronize with the received signal.

## 3. Real-Time Signal Processing Implementation

The previously described demodulation blocks—BPF1, FM DEM, and BPF2—were implemented in MATLAB for the purposes of concept development and testing. However, this implementation is not suitable for real-time applications. To enable real-time processing, the appropriate code and algorithms must be ported to a real-time environment, ideally one that is open source and platform-independent. For this reason, a real-time solution using JavaScript, HTML, and CSS was chosen, with the implementation logic illustrated in Figure 5.

Many of the functional blocks, such as AS, BPF1, ScriptPreprocessorNode, and AudioWorklet, are components of the Web Audio API—a powerful JavaScript API that enables audio manipulation and analysis directly in the browser. The Web Audio API is part of the modern web platform and is widely supported across most browsers, including Chrome, Firefox, Safari, and Edge.

The functional blocks are utilized from the Web Audio API according to the following signal flow logic [16,17,18]:

source → node1 → node2 → … →nodek →
→custom audio processing script → destination [19].

The FM DEM, BPF2, and visualization blocks are implemented within a custom audio processing script, which is activated upon receiving an Audio Processing Event (APE) from the Web Audio API. The ScriptProcessorNode and AudioWorklet are employed to enable direct, synchronized access to input data, allowing for real-time, frame-by-frame manipulation with precise timing. The commonly used AnalyserNode from the Web Audio API cannot synchronize the process and is therefore not applicable in this scenario.

It is important to note that ScriptProcessorNode operates on the main thread and may experience glitches if the user interface is busy, making it unsuitable for high-precision, real-time audio processing. In contrast, AudioWorklet runs on its own thread, avoiding such glitches and providing smooth and reliable audio processing. For these reasons, AudioWorklet is strongly recommended for real-time applications requiring high precision [20].

Unlike MATLAB, the processing blocks BPF1, FM DEM, and BPF2 are implemented entirely in the time domain in the web-based approach. BPF1 utilizes built-in Web Audio API functions, while FM DEM and BPF2 are custom-implemented in JavaScript. Figure 6 and Figure 7 illustrate the real-time methodology for processing medical data over sound using the Web Audio API and JavaScript code.

Figure 6 shows the scenario in which the analog ECG signal, ECG(t), is transmitted over sound. The upper diagram presents an experimental spectrogram recording obtained using the Spectrogram application [21], with the following parameters: sampling rate = 48 kHz, FFT size = 1024 bins (47 Hz/bin), decimation = 5 (1.5 Hz/bin at DC), window function = Hamming, desired transform interval = 10 ms (100 Hz), and exponential smoothing factor = 0.1. The lower left panel displays the digitized signal $V_{m}\left(t\right)$ in the form of $V_{1}\left(n\right)$ after preprocessing with BPF1, implemented in JavaScript using the Web Audio API, with the following filter parameters: BPF1 = [15 kHz, 20 kHz], BPF2 = [0.5, 15] Hz, ZC demodulation, and gain = 100. The lower right panel shows the reconstructed, decoded ECG signal $V^{\prime }\left(nT_{s}\right)=V^{\prime }\left(n\right)$.

Figure 7 presents a similar situation, where the ECG(t) signal is replaced by a digital value or character—for example, the ASCII character ‘A’ in UART format, with a baud rate of 10 bits/s. The filter parameters are as follows: BPF1 = [15 kHz, 20 kHz], BPF2 = [0.1, 50] Hz, ZC demodulation, and gain = 100. The spectrogram parameters remain the same as described above.

## 4. Verification and Testing

In this phase of the development and piloting of the CardiaWhisper device, the design methodology and preliminary results are evaluated according to the following criteria: the functionality of the underlying principle, the selection of suitable software demodulation algorithms for platform-independent real-time applications, power consumption, and the communication range.

### 4.1. Functionality

The functionality of the system was evaluated using the experimental setup shown in Figure 8. The ECG signal, acquired with a standard three-electrode configuration, is captured by the CardiaWhisper device prototype (1) and transmitted to multiple mobile phones and desktop computers, following the general scenario outlined in Figure 1. The electrodes used are of the standard type found in Holter or external loop recorders, capable of continuous 24–72 h recording. These are disposable, adhesive, gel-based Ag/AgCl (silver/silver chloride) surface electrodes. In some experiments, clamp electrodes for the extremities (hands and legs) were also used.

The CardiaWhisper device consists of an ECG front-end based on the AD8232 chip [22], an FM modulator built with a CD4046 VCO [23], and a piezo speaker with appropriate power and spectral response. The transmitter is powered by a 9V battery. Additionally, a transmitter running on Arduino NANO was used in tests, particularly for digital signals and ECG signals with reduced resolution.

A control and monitoring mobile phone (2), running the Spectroid Application [21], displays the FFT and STFT spectra of the transmitted ECG signal. This allows the verification of system parameters and communication quality in both indoor and outdoor environments. A second mobile phone (3) is used to run custom-developed JavaScript software for demodulation and visualization, originally developed in Visual Studio Code Version 1.98.2.

Testing was conducted in environments ranging from quiet rooms (30–40 dB) to urban streets (60–70 dB) and normal restaurants (up to 80 dB). Environments exceeding 80 dB were not considered; however, at close proximity (within 10 cm), the system remains functional even in noise levels up to 100 dB.

Users access the application via a web page hosted on a company server, launching the CardiaWhisper software (JavaScript+HTML+CSS) to receive and display the “over-the-air” ECG signal. The desktop computer (4) simultaneously receives the same ECG signal via its built-in microphone and is used for software development, parameter adjustment, testing, and debugging. This experimental setup fully demonstrates the functionality of the CardiaWhisper approach. The JavaScript+HTML+CSS application operates in real time on modestly equipped phones, including older Android and iPhone models such as the Samsung Galaxy J4. ECG signals were sourced both from a simulator and from a group of volunteers.

Figure 9 presents real-time, JavaScript-based demodulation of an ECG signal from a 58-year-old male volunteer, using the CardiaWhisper device and Google Chrome browser decoding software with the following parameters: BPF1 = [15 kHz–20 kHz], BPF2 = [0.5–15 Hz], ZC demodulation, and gain = 1. Both the modulated and demodulated signals are illustrated.

### 4.2. Signal Quality and SNR Measurements in Indoor Environments

Signal reconstruction quality depends on several factors, including transmitter power, receiver distance, receiver sensitivity, the presence and type of obstacles, wall reflection, and the performance of the reconstruction (demodulation) algorithm. In this phase of the research, we evaluated signal quality at the receiver location by calculating the signal-to-noise ratio (SNR) as follows:(10)$${\mathrm{SNR}}_{\mathrm{dB}}=20\log_{10}\left(\frac{V_{\mathrm{RMS}}\left(\mathrm{signal}+\mathrm{noise}\right)}{V_{\mathrm{RMS}}\left(\mathrm{noise}\right)}\right)$$
where $V_{\mathrm{RMS}}$ (Root Mean Square) expresses the effective value of the received signal at the point of measurement.

The experimental setup is depicted in Figure 10a. In the indoor area, the transmitter (TX, 10 mW output) emits a frequency-modulated (FM) sound signal with an 18 kHz carrier and a 1 Hz, 1 V_pp_ sinusoidal modulating signal. This signal is detected and analyzed by a microphone connected to a Keuwlsoft spectral analyzer application [24]. The receiver (RX) was placed at various locations inside the premises: in direct line of sight, behind an obstacle, behind an ajar door, and outside (with an open door). The distance *d* represents the straight-line separation between the transmitter and the receiver. Environmental loudness was varied from silent to loud room conditions.

Figure 10b shows the measured SNR as a function of distance for different scenarios. The accepted thresholds for signal quality were defined as follows: VERY GOOD (SNR≥30 dB), GOOD (20dB<SNR<30 dB), and FAIR (10dB<SNR<20 dB).

As can be observed, even under “Loud + obstacles” conditions, VERY GOOD signal quality (SNR≥30 dB) is achieved within a radius of d=1.5 m from the TX, and GOOD quality is maintained up to d=4.2 m. These results demonstrate the robustness of the system in real-world indoor environments.

### 4.3. Platform-Independent Implementation

After implementation and testing in MATLAB, the code was translated to JavaScript + HTML + CSS, which serves as a platform-independent solution thanks to its cross-browser compatibility and the “write once, run anywhere” (WORA) paradigm. The zero crossing (ZC), derivative-based (DIFF), and Hilbert-transform-based demodulation algorithms were evaluated in terms of signal reproduction quality, noise resistance, and processing speed.

Figure 11 illustrates representative results for both “calm” environments and noisy environments with background speech or music. As shown, ZC demodulation provides satisfactory results in both cases. The DIFF method also performs well but is less immune to signal transitions, artifacts, and noise. The Hilbert-based demodulation, however, yields the highest quality of signal reconstruction, with superior resolution and noise resistance.

In terms of speed, as shown in Table 1, ZC is the fastest method, followed by DIFF, while the Hilbert implementation is significantly slower due to its computational complexity. Speed testing was conducted on an Intel(R) Core(TM) i5-5350U CPU @ 1.80 GHz, 1801 MHz, with two cores and four logical processors, and 8.00 GB of installed physical memory (RAM). A 10.92-s signal was sampled at a frequency of 48,000 Hz, resulting in 524,160 samples, with tests repeated over 30 trials.

Table 2 summarizes the comparison of four demodulation techniques—HIL1 (Hilbert-based), HIL2 (Hilbert-based), ZC (zero crossing), and DIFF (Slope-based)—evaluating each method based on complexity, implementation suitability, noise immunity, artifact resistance, and real-time suitability.

### 4.4. Power Consumption

The CardiaWhisper is a low-power device. The ECG signal is acquired using a low-cost ECG amplifier based on the AD8232 chip [22] and is subsequently modulated by a micro-power VCO modulator based on the CD4046B [23], with both components powered at 3.3V. The microcontroller-based modulator utilizes the ATMEGA328P chip, operating at 16MHz and powered at 5V. The required working voltages for the modulators are derived from a 9V battery using a low-dropout (LDO) regulator. The overall power consumption of the transmitter for different configurations is summarized in Table 3.

These results are comparable to those of standard Bluetooth Low Energy (BTLE) microcontrollers, which typically consume approximately 30mW in active mode during real-time signal streaming. The measured consumption is somewhat higher than that of ultra-low-power configurations, such as the nRF52 family.

### 4.5. Range

CardiaWhisper has been tested primarily in indoor environments. For reference, a sensitivity of −16dB was measured at a distance of 10cm from the emitter (speaker), with 0dB corresponding to a full-scale ±1V audio signal, which is difficult to achieve in air with an audio codec gain of 1. Satisfactory signal reconstruction was achieved down to −40dB in indoor use, corresponding to approximately 5m in open space and about 2.5m in a furnished environment. Reliable operation was also observed through walls at distances up to 3m. It should be noted that the achievable range depends on multiple parameters, and CardiaWhisper is generally not intended for distances greater than 1.5m. The simplest way to increase the sensitivity of CardiaWhisper is to adjust the gain of the microphone amplifier within the audio codec; in some applications, gains of up to 200 were used.

## 5. Discussion

Through work on the CardiaWhisper project, the authors have gained valuable insights into the use of sound as a medium for transmitting medical data. Key findings include, but are not limited to, the following:The classic concept of “data over sound” can be extended to include medical data over sound, or indeed any sensor data over sound. Data may be analog, digital, or mixed.Air, fluids, and wires can all serve as transmission media, although the effective range is typically limited to around 1.5–3m indoors, depending on transmitter power and receiver sensitivity.Transmission of analog signals supports 2–3 channels, while digital transmission allows up to 4 channels, albeit at low data rates (tens of bits per second).One-way communication is preferred, although duplex communication is possible.Transmitters can be simple piezo speakers powered by basic circuits, and receivers can be inexpensive and based on widely available microphones (e.g., electret). MEMS microphones utilizing pulse density modulation (PDM) are currently very effective and promising.The system is subject to noise from ambient sound, physical obstacles, movement, and, rarely, electrical interference.Very low power is required: transmitters operate with only a few milliwatts, and receivers can function with low power consumption—modern MEMS microphones can operate in the tens of microwatts to milliwatt range, depending on the implementation.Signal encoding is typically implemented via frequency or phase modulation; amplitude modulation is impractical in air but viable over wire. Square waves may be used in place of sinusoids for simplicity.Receivers can employ software-based filter banks and real-time demodulation methods, ranging from basic zero-crossing detection to more advanced techniques.The system is safe for health at low power levels.Applications in medicine are numerous, including but not limited to-Passive sensing (e.g., stethoscope, cough detection);-In-body ultrasound communication;-Communication for setting or reading implant devices;-Near-field alerting (e.g., sleep apnea detection, fall alerts);-Local broadcasting of events (e.g., arrhythmia alerts, rhythm disorder notifications);-Vital sign visualization (ECG, PPG, ACC) in time, frequency, and time-frequency domains;-Haptic signaling via smart wearables (e.g., alerting nearby devices using a speaker in clothing);-Zero-configuration monitoring of vital signs via web browser interfaces, accessible from any device;-Fall and presence detection;-etc.

While EM-based communication is generally more robust and broadly applicable, sound-based transmission offers unique advantages for low-power, short-range, and privacy-sensitive medical use cases, all under certain conditions.

## 6. Conclusions

This paper has introduced an alternative methodology for near-field communication in medical sensor devices, utilizing transmission within the near-ultrasound (audio/acoustic) range. Building upon the established concept of data over sound (DoS), which has gained renewed attention across various applications, we extend the paradigm to medical data over sound (MDoS). As a case study, the CardiaWhisper system demonstrates the real-time transmission of cardiological (ECG) data over sound.

The proposed approach encodes analog ECG signals, acquired through standard loop recorder or Holter configurations, and transmits them acoustically to nearby consumer devices—such as mobile phones, tablets, or desktop computers—equipped only with built-in microphones. These signals can then be visualized, decoded, analyzed, and logged using open-source, platform-independent software, with the option to forward data to local or global networks. The system’s architecture—including both transmitter and receiver components—was described, with particular emphasis on the receiver side, where several software-based demodulation algorithms were implemented using HTML, JavaScript, and CSS.

Preliminary results have been presented, compared, and discussed, showing that the CardiaWhisper system achieves reliable data transfer, robust real-time demodulation, and satisfactory signal quality under typical indoor conditions. The advantages and limitations of the medical-data-over-sound methodology have also been discussed, including its low power consumption, platform independence, and EMI resistance, alongside challenges such as limited range and sensitivity to acoustic noise.

The proposed approach opens new possibilities for low-power, privacy-conscious, and easily deployable medical sensing and communication, with potential applications ranging from home health monitoring to telemedicine and implant communication. Future work will focus on optimizing data rates, expanding use cases, and further improving noise immunity and system robustness.

## Figures and Tables

**Figure 1 sensors-25-04573-f001:**
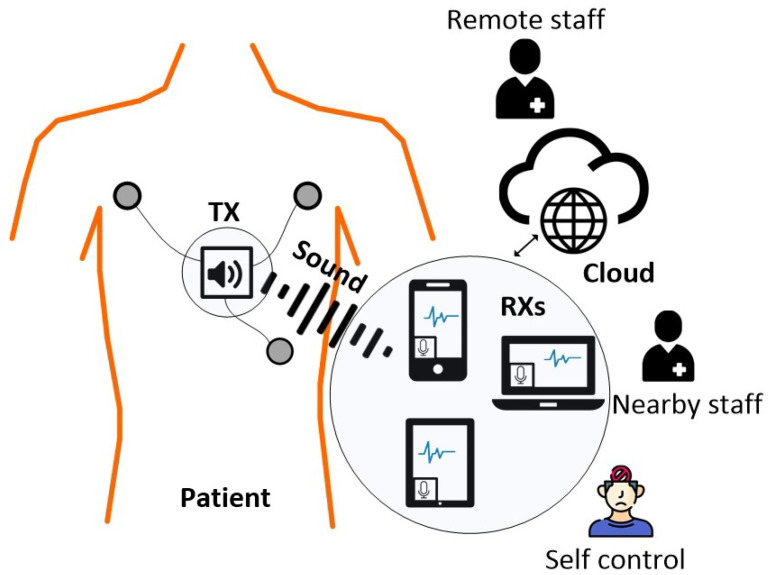
“CardiaWhisper” general architecture. Multiple devices can simultaneously capture the real-time ECG signal through their built-in microphones, enabling monitoring by staff, cloud platforms, and the patient.

**Figure 2 sensors-25-04573-f002:**
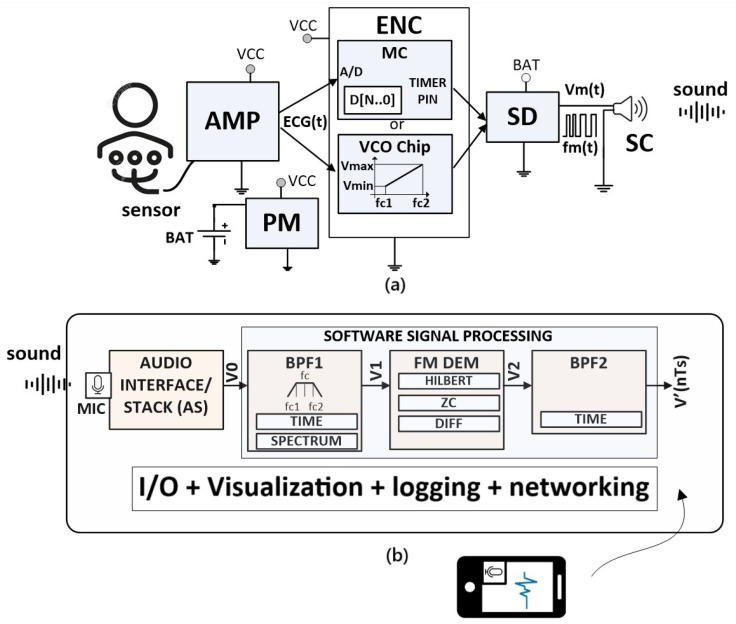
“CardiaWhisper” multi-receiver architecture, with (**a**) showing the transmitter block diagram for signal amplification, encoding, and acoustic transmission, and (**b**) illustrating the receiver block diagram for signal acquisition, digitization, and software-based processing on general devices.

**Figure 3 sensors-25-04573-f003:**
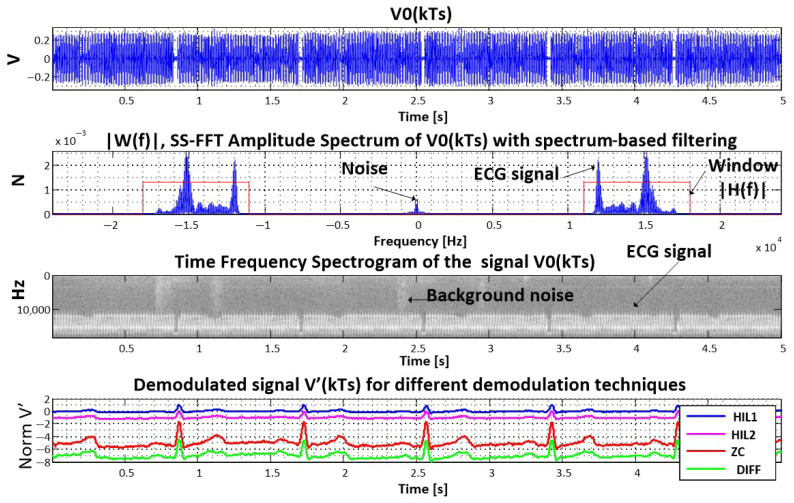
Illustration of software FM demodulation for medical data over sound, showing the time signal, spectrum-based filtering, spectrogram, and comparison of Hilbert transform, zero crossing, and derivative-based demodulation techniques.

**Figure 4 sensors-25-04573-f004:**
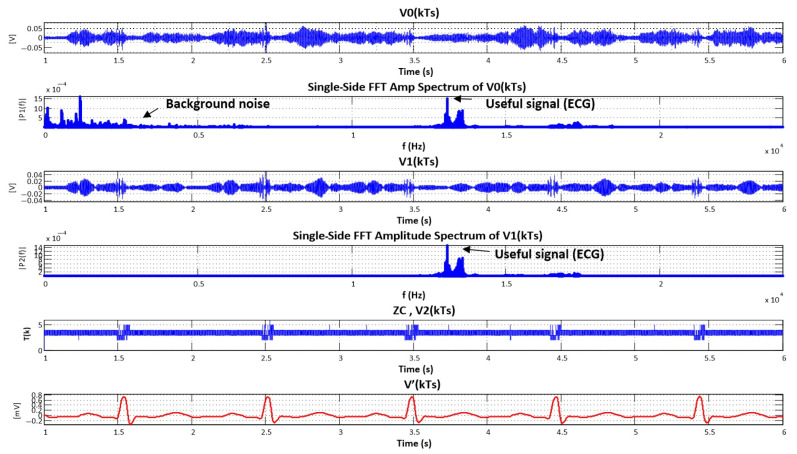
Illustration of software FM demodulation steps using the zero-crossing (ZC) technique, showing time-domain signals, spectral filtering, ZC extraction, and the resulting demodulated ECG signal.

**Figure 5 sensors-25-04573-f005:**
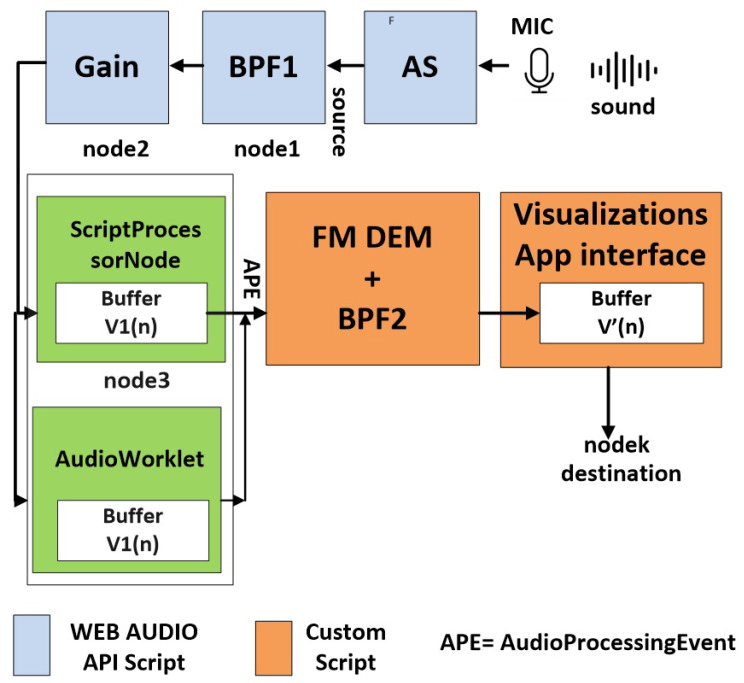
A block diagram of real-time FM demodulation and visualization in a web browser, implemented using the Web Audio API and custom JavaScript scripts. Signal flow includes audio acquisition, filtering, demodulation, and real-time visualization using ScriptProcessorNode and AudioWorklet.

**Figure 6 sensors-25-04573-f006:**
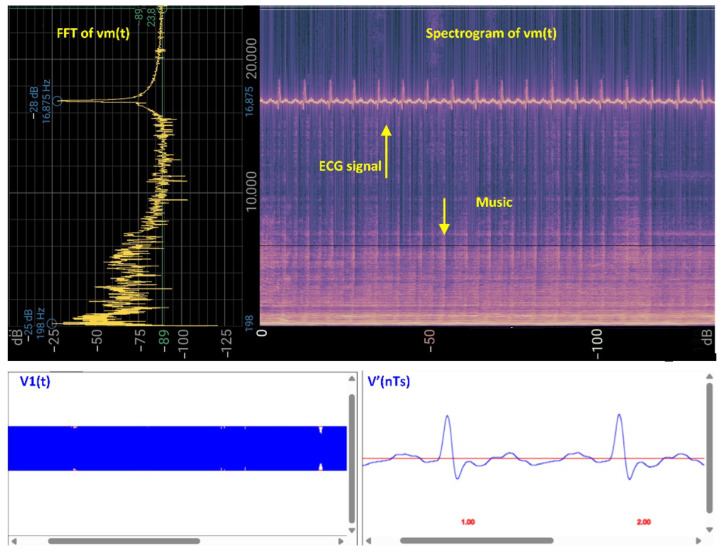
Illustration of the main audio signal processing workflow implemented in the CardiaWhisper concept for the transmission of an analog ECG signal over sound. (**Top left**) FFT of the audio signal $v_{m}\left(t\right)$, showing the frequency spectrum. (**Top right**) A spectrogram of $v_{m}\left(t\right)$, where the embedded ECG signal occupies the 16–20kHz frequency range. The music signal is present at lower frequencies and does not interfere with the transmitted ECG signal. (**Bottom left**) The digitized signal $v_{1}\left(n\right)$ at the receiver side, after preprocessing with the first band-pass filter (BPF1). (**Bottom right**) The reconstructed and decoded ECG signal $V^{\prime }\left(nT_{s}\right)=V^{\prime }\left(n\right)$, as obtained from the demodulation and postprocessing steps. Both $v_{1}\left(n\right)$ and $V^{\prime }\left(n\right)$ are obtained in real time within a web browser environment using JavaScript, HTML, and CSS.

**Figure 7 sensors-25-04573-f007:**
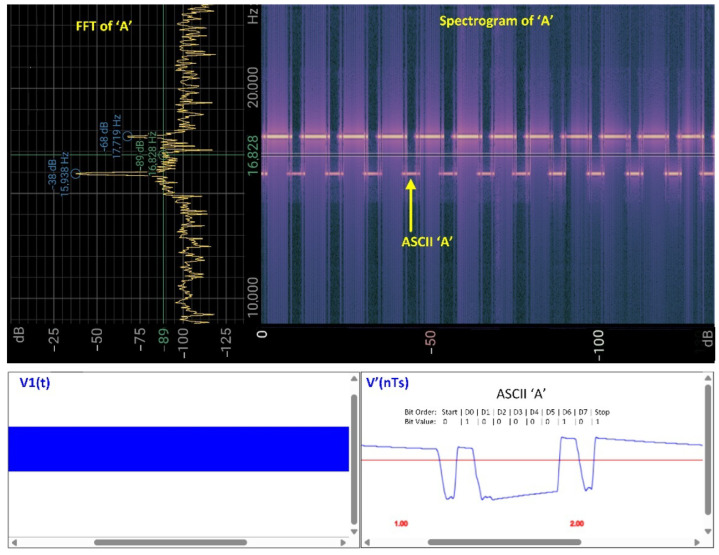
Illustration of digital signal transmission in the CardiaWhisper system, referencing the scenario shown in Figure 6. In this case, instead of transmitting the analog ECG(t) signal, a digital signal D[7..0], corresponding to the ASCII character ‘A’, is sent over sound. (**Top left**) FFT of the audio signal representing ASCII ‘A’. (**Top right**) A spectrogram of the received signal, where the sequence corresponding to ASCII ‘A’ is visible in the designated frequency range. (**Bottom left**) The digitized signal $v_{1}\left(n\right)$ at the receiver after preprocessing with BPF1. (**Bottom right**) The reconstructed digital signal $V^{\prime }\left(nT_{s}\right)$, where the bit values and order (Start, D0–D7, Stop) corresponding to the ASCII ‘A’ character are clearly shown. As before, the processing and visualization are performed in real time within a web browser using JavaScript, HTML, and CSS.

**Figure 8 sensors-25-04573-f008:**
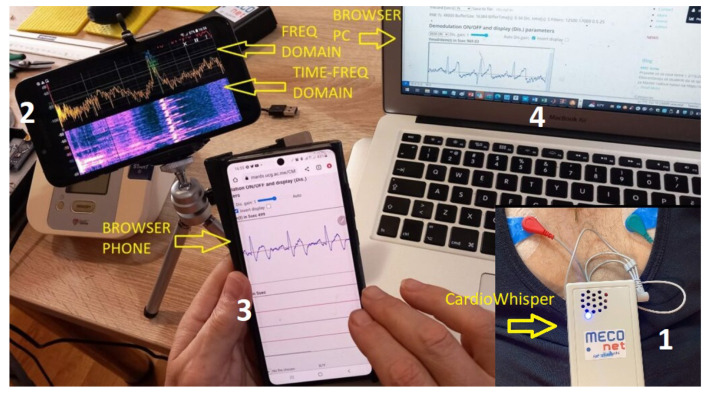
The experimental setup demonstrating the CardiaWhisper system. (1) The CardiaWhisper device acquires ECG signals using standard disposable Ag/AgCl electrodes and transmits data via sound. (2) A mobile phone displays the signal’s spectral content for system verification. (3) Another phone runs custom JavaScript software for real-time demodulation and ECG visualization. (4) A desktop computer receives and analyzes the same signal simultaneously for development, testing, and debugging purposes.

**Figure 9 sensors-25-04573-f009:**
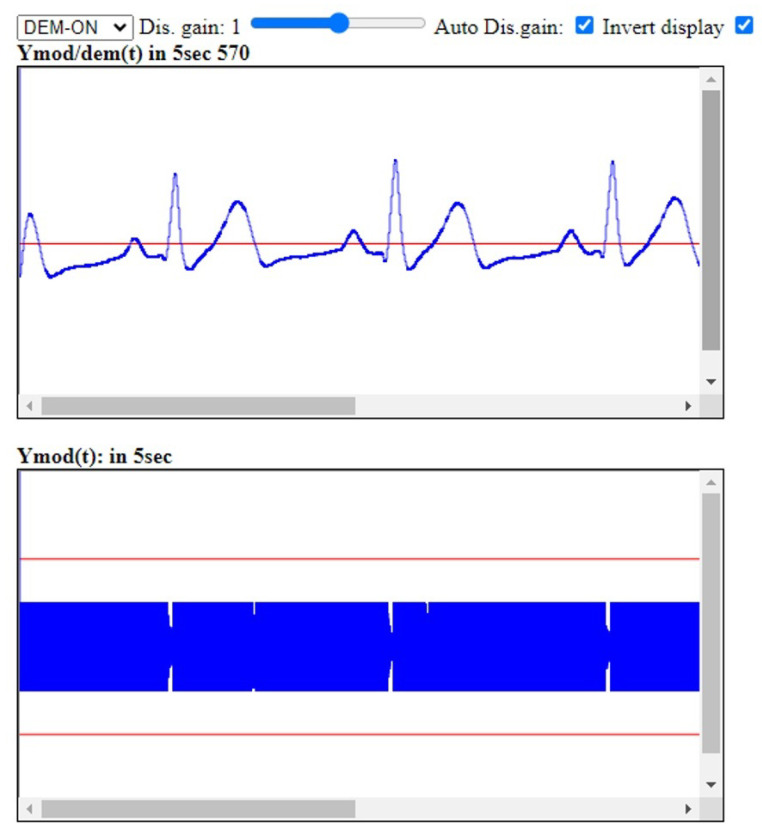
JavaScript-based software FM demodulation in the time domain using the zero-crossing (ZC) technique. (**Top**) The demodulated ECG signal at the receiver side. (**Bottom**) The FM-modulated ECG signal $y_{\mathrm{mod}}\left(t\right)$ as received. The full window in the *x*-direction corresponds to 5s, while the *y*-axis range of ±1V for the audio signal corresponds to ±1mV of the original ECG signal.

**Figure 10 sensors-25-04573-f010:**
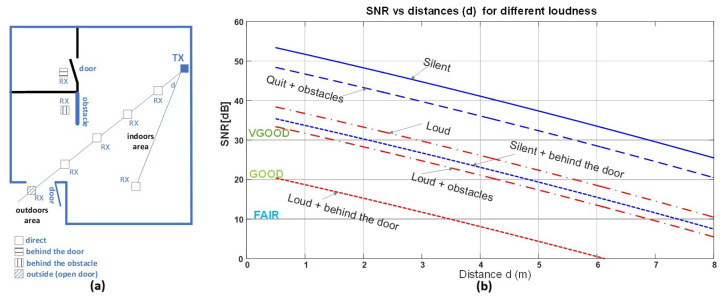
(**a**) The floor plan and measurement scenario: The transmitter (TX) is placed indoors, while receivers (RX) are positioned at various locations—direct line of sight, behind obstacles, behind a door, and outdoors (with an open door). (**b**) The signal-to-noise ratio (SNR) as a function of distance *d* for different environmental loudness conditions and receiver positions.

**Figure 11 sensors-25-04573-f011:**
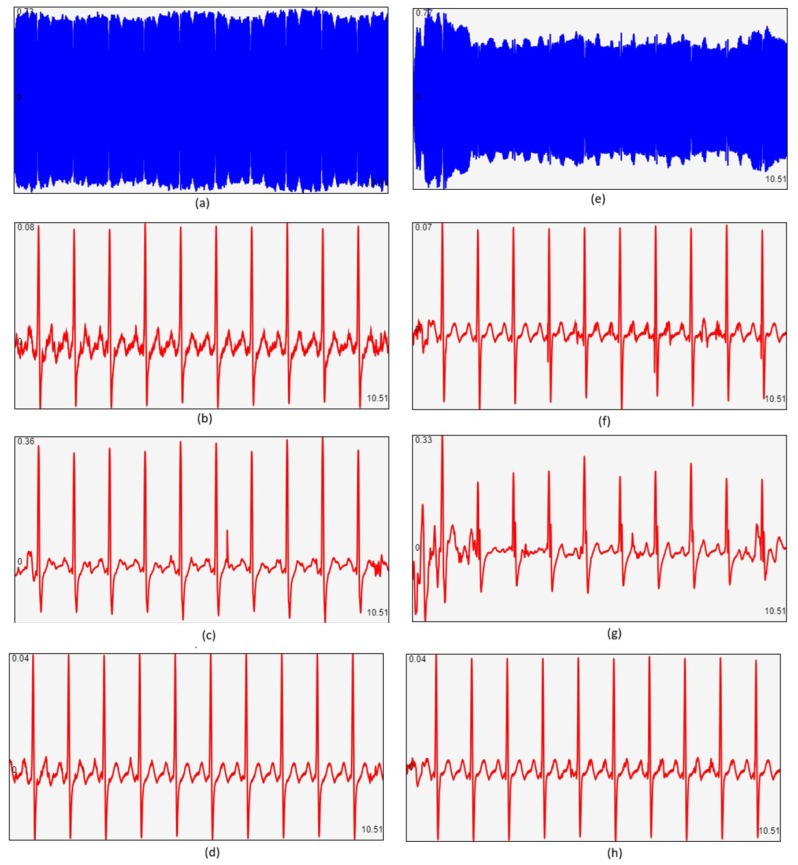
Results of real-time hardware implementation of FM demodulation algorithms. (**a**) FM-modulated ECG signal in a calm environment; (**b**) demodulation using the zero-crossing (ZC) technique; (**c**) demodulation using the derivative-based (DIFF) technique; (**d**) demodulation using the Hilbert-transform-based (HIL2) technique. (**e**) FM-modulated ECG signal in a “noisy” environment; (**f**) ZC-based demodulation; (**g**) DIFF-based demodulation; (**h**) Hilbert-based (HIL2) demodulation. Each plot shows a 10.5s window of the ECG signal.

**Table 1 sensors-25-04573-t001:** Speed test results.

Method	ZC	DIFF	Hilbert (HIL2)
Average speed in ms for processing 524,160 length Float32 array	16.7	22.4	760.5

**Table 2 sensors-25-04573-t002:** Comparison of demodulation techniques.

Demodulation Technique	Complexity	Implementation Suitability	Noise Immunity	Artifact Resistance	Real-Time Suitability
HIL1 (Hilbert-based)	High	Complex in JavaScript, feasible in Python and MATLAB	High	High	Suitable
HIL2 (Hilbert-based)	High	Complex in JavaScript, feasible in Python and MATLAB	High	High	Suitable
ZC (Zero Crossing)	Low	Simple and suitable for JavaScript	Moderate	Moderate	Suitable
DIFF (Slope-based)	Low	Moderate complexity; less suitable due to noise and artifacts	Low	Low	Suitable

**Table 3 sensors-25-04573-t003:** Power consumption of the CardiaWhisper transmitter in different modulator configurations.

Configuration	VCO 4046-Based Modulator	ATMEGA328-Based Modulator
3.3 V_pp_ on piezo speaker	15.5 mW	34 mW
5 V_pp_ on piezo speaker	20 mW	70 mW
9 V_pp_ on piezo speaker	35 mW	90 mW

## Data Availability

The original contributions presented in this study are included in the article. Further inquiries can be directed to the corresponding author.

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
