# Peer review of "Medical Data over Sound—CardiaWhisper Concept"

_sensors, 2025, doi:10.3390/s25154573_

Round 1

Reviewer 1 Report

Comments and Suggestions for Authors

The manuscript has severl issues that need to be addressed. The abstract doesn't really emphasize the practical aplication and contributions of the system, especially in remote healthcare and real-time health monitioring. The background lacks a more detailed discussion of the challenges faced by current wireless communication technologies in healthcare, especially with electromagnetic interference and power consumption. In the methodology section, they don’t explain very well why they picked specific signal modulation and demodulation techniques (like ZC, DIFF, Hilbert), and the limitations and assumptions of the experiment setup are missing. The results section doesn't have any actual performance metrics like SNR (Signal-to-Noise Ratio) or PSNR (Peak Signal-to-Noise Ratio), and the comparison between different demodulation techniques lacks deeper analysis of trade-offs in real-world applications where noise and interference can vary. The future work section is a bit vague and should lay out more specific plans to address system limits, like speeding up real-time processing or expanding the system to handle other bio signals. Also, some figures don't have good explanations in their captions, and certain figures, especially the ones comparing different demodulation methods, could be made clearer. The manuscript also uses some technical terms that non-expert readers might have trouble with, and some parts of the methodology and results discussion are too technical and could be easier to understand or more concise. Finally, the experiment setup needs more detail, especially when it comes to noise conditions and test scenarios, like distances, noise levels, and device setups, to make sure the results are reproducible.

Comments on the Quality of English Language

The quality of the English in the manuscript is okay overall, but there’s a few places where it could be better. Some of the technical terms and ideas aren't explained well enough, which could make it hard for people who aren’t experts to understand. In some sections, especially the methodology and results parts, the language is way too technical, so it might be hard for a wider audience to follow. Also, the sentence structure could be a bit simpler in some places to make it easier to read and clearer. There’s also some spots where the terminology isn’t used consistently, and it could definitely use a little more proofreading to make sure everything’s accurate and flows well. All in all, the English works, but making it clearer, more consistent, and easier to read would really help make the manuscript better.

Reviewer 2 Report

Comments and Suggestions for Authors

The author has proposed a CardioWhisper transmission technology to connect health monitoring devices with terminals such as smartphones. This is a practical signal transmission technology, which is a valuable contribution to the field. The paper is well organized and written. I have several comments:

1.     It would be helpful if the author could provide a comparison of the advantages and disadvantages of this technology in relation to Bluetooth technology.

2.     The resolution of certain images is insufficient, such as Figures 9 and 10.

3.     The author should compare different signal reconstruction methods in a quantitative manner, such as providing accuracy data.

4.     The author's description of the power consumption section is too brief and needs to be more detailed.

Reviewer 3 Report

Comments and Suggestions for Authors

Manuscript: sensors-3446685

The present manuscript entitled “CardioWhisper – A Low-Power, Near-Ultrasound Communication System for Medical Wearables in IoT and Edge Applications” has been written in a systematic manner distinguishing the key features of the present state of art. However, the manuscript can be accepted after entertaining the following queries raised by reviewer:

Q1. The author claims that the “Pilot tests highlight "CardioWhisper’s" noise resilience, low power consumption, speed, and reliability, reinforcing its suitability as a cost-effective, user-friendly solution for telemedicine, home care, and telemedical applications” in line 9-11. It would be more effective if the authors could include one table comparing these properties w.r.t., the available wearable gadgets.   

Q2. The authors didn’t explain regarding the material that have been employed for the electrode materials. Is it natural and eco-friendly? Also, how did they fixed on the subject body?   

 Q3. In Figure 10, the authors have shown the transient behavior Voltage vs Time. However, it’s difficult to understand how much change in the magnitude observed during different cases. Therefore, I recommend using scale along the X- and Y- axis.

Q4. I am wondering if the authors have recorded the response using different subjects (patients) or at different stages of activities (normal walk, brisk walk or running) to compare their device performance in actual situation.

Reviewer 4 Report

Comments and Suggestions for Authors

The article describes a communication system based on FM modulation of a relatively high-frequency sound (in the 16-20 kHz) range used to transmit the signal obtained by a portable ECG detector. The reception is software-based demodulation in several devices (phones, laptops) through standard microphones in those devices.

Although the application is an interesting exercise and can have its points of merit, I’d like to see more emphasis, in general, on a series of points that have been underdeveloped in the study.

  1. I guess that the frequency has been chosen so that most people cannot hear it (line 51 needs a reference), and it still passes through the standard anti-aliasing filter of audio devices, typically around 20 to 22 kHz. But some people do hear it (babies, very young children). Has it been tested on them?

  2. The transmitter is described as having “sufficient power” (line 183); you need to specify more, which can also be related to the previous point.

  3. The test has been done with external noise/music, but a much more important source of noise at those frequencies, especially in a home environment, are kitchen and utility devices (food mixers --- try it) that use PWM-based motors. You should check this and add some number on (acoustical) noise robustness in terms of SNR levels.

  4. The power consumption comparison (4.2, line 222 and ff) needs more numbers. Transmitting an ECG with BTLE could involve sending about 200 bytes every second, permitting a stand-by of more than 99% at typical baud rates, so you should specify more the details in which the comparison is done.

  5. It seems clear that only one such device can be used at a time, in the sense that you need to have just one measurement per household and that interferences between several CardiWhispers will be a fact in the outside environment. Could you reduce the band used to solve that? Maybe fitting four to eight channels shared in the 16-20 kHz band or some kind of digital channel codification? That could also help in the clear lack of privacy concerns for this kind of communication and the easiness of tampering with it.

  6. You should also compare this system with the commonly available chest bands and the widely used Holter systems.

Apart from the previous major points, several minor ones should be corrected.

  1. In line 53, what do you mean by “utilizing natural voice”?

  2. In lines 57-58, there are no proven harmful effects of non-ionizing EM waves; remove or justify with sound references.

  3. Figures 1 and 2 should be compacted into one.

  4. In the receiver, you should comment somewhere about the anti-aliasing used (even if using the standard audio one).

  5. Figure 5 is the only one with a two-sided spectrogram — please unify.

  6. The blue blob of Figure 9 (down) is of little usefulness. Show a time-zoomed part of it so that something can be seen. A similar (but less jarring) happens in Fig. 10.

Reviewer 5 Report

Comments and Suggestions for Authors

Authors presents CardioWhisper, an acoustic based communication system for transmitting biomedical signals from wearable sensors to mobile devices using near-ultrasound frequencies (16-20 kHz). The authors claim that this method provides low power consumption, minimal EMI, and compatibility with existing microphones in consumer devices (cellphone, laptops and etc). The system implements FM modulation for signal transmission and real-time JavaScript-based demodulation, enabling platform-independent signal processing. Testing was conducted to evaluate power efficiency, communication range, and real-time performance.

This reviewer believes the manuscript needs major revisions before being suitable for publication. Mainly, this manuscript lacks quantifiable measures for performance of this system.

1. Authors fail to clearly articulate what is novel about this system. Ultrasound communication in biosensors is well established, and the authors need to explicitly state how their system differentiates itself from state of the art. What specific advantages does CardioWhisper provide over other acoustic or RF-based methods?
2. Authors primarily made power consumption comparison with BLE technology and fail to provide citation and comprehensive comparison against similar acoustic (or ultrasonic) communication methods. While low power consumption is a strength, range is also a key limitation that must be properly evaluated. A more comprehensive trade-off of all KPIs is necessary.
3. Authors claim low power consumption but fail to quantify the acoustic power (in dB).
4. What is the measured SPL (Sound Pressure Level) at different distances?
5. Given that 16-20 kHz is within human hearing range (although for young individuals), they must discuss potential health effects (headache, tinnitus, nausea, etc.) vs transmitted power and SPL.
6. Acoustic communication is susceptible to background noise (e.g., human speech, music, HVAC systems, machinery). How does CardioWhisper handle external noise sources? Authors need to quantify the noise level immunity.
7. What is the Signal-to-Noise Ratio (SNR) to achieve "satisfactory signal quality"?
8. The manuscript lacks quantification of different demodulation and filtering techniques. Authors should provide performance comparisons in terms of accuracy, noise immunity, and computational efficiency.
9. The theoretical discussions about FM, FM demodulation, and filtering methods are redundant. These methods are well established.
10. The term "Near-Ultrasonic" is misleading. A better term might be "high-frequency audible acoustic communication."
11. What is the effect of the receiver’s microphone (bandwidth, sensitivity vs. frequency) on the performance of the system?

Round 2

Reviewer 1 Report

Comments and Suggestions for Authors

The authors have revised the manuscript substantially and addressed many of the earlier comments. The presentation is clearer, and the technical implementation is more solid. Before accepting the manuscript, I believe there are still improvement. 
The manuscript still does not provide quantitative measures  to objectively evaluate signal quality. Without such metrics, it is difficult to assess the robustness of different demodulation methods.

Although processing speed and general noise immunity are now discussed, the comparison remains largely qualitative. It would strengthen the work to include representative test scenarios more aligned with pratical cases—e.g., dynamic noise environments, user movement, or varying source-receiver distances.

Reviewer 4 Report

Comments and Suggestions for Authors

The authors addressed most of the concerns in my first review; the only (relative) problem I find is that the article has been changed so much that it should have been a resubmission, not just an (albeit major) revision.

The only suggestion is to give a more easily readable description of the filter used: the filter in equation 4 is obviously non-causal, so the implementation will be an approximation of it;  so, which kind of band-pass has been implemented? This is critical for recovering a signal in the range 16-20 kHz sampled at only 48 kHz (just a bit over the Nyquist's limit). 
